## Research Article

interdisciplinary research; institutional barriers; early-career researchers; career progression; systemic challenges

**Corresponding author:**
Marta Payo Payo;
Email: marpay@noc.ac.uk

# Navigating interdisciplinary coastal research in the UK: Challenges and solutions from an early career perspective

Elina Apine[1] 📷, Marta Payo Payo[2] 📷, Amani Becker[2], Marta Meschini[3], Constantinos Matsoukis[2] and Sara Kaffashi[4,5]

[1]School of Geography and Sustainable Development, University of St Andrews, St Andrews, UK; [2]National Oceanography Centre, Liverpool, UK; [3]School of Environmental Sciences, University of Liverpool, Liverpool, UK; [4]Economic Consultant, Oxford, UK and [5]Cranfield Environment Centre, Cranfield University, Cranfield, UK

## Abstract

Coastal areas are vital hubs for diverse ecosystems and socio-economic activities, but they face significant threats from climate change, biodiversity loss and pollution. These challenges require urgent, cooperative actions and interdisciplinary approaches to develop sustainable solutions. However, interdisciplinarity requires blurring traditional academic disciplinary boundaries, and this can be a challenge. Increasingly, early-career researchers (ECRs) are undertaking interdisciplinary research while facing uncertainty about their career progression. In this research paper, we explore the challenges and opportunities faced by ECRs in the United Kingdom conducting Interdisciplinary Coastal Research (IDCR). We draw on findings from internal workshops, webinar discussions and an online survey, all conducted in 2024. The main barriers to IDCR are systemic in nature and include demanding workload, short-term contracts, ineffective supervisory and limited institutional support. Generally, ECRs felt positive about the benefits of interdisciplinarity to coastal research and their career development, but some ECRs expressed feelings of impostor syndrome. Enhanced flexibility in approaches, improved communication and open-mindedness are among the proposed solutions. This research highlights the mismatch between the ambition and the day-to-day reality of ECRs working in IDCR and provides recommendations for IDCR, which can both enhance the experience of ECRs and secure better outcomes for coastal areas.

## Impact statement

The impacts of the triple planetary crisis of biodiversity loss, climate change and pollution are particularly felt at the coast. The need for interdisciplinary approaches is increasingly acknowledged by major United Nations (UN) programmes, such as the UN Ocean Decade and the UN Sustainable Development Goals. In the United Kingdom, interdisciplinarity is a goal in itself for UK Research and Innovation-funded programmes, encouraging systems approaches to tackle problems identified by policymakers, and to increase national capability in inter- and transdisciplinary research. However, our research into Interdisciplinary Coastal Research (IDCR) in the United Kingdom identifies a gap between the high-level institutional ambition and the experience of early-career researchers (ECRs), who undertake a substantial proportion of the work.

Our findings align with the existing literature, offering additional evidence to support the highlighted barriers, including demanding workload, limited institutional support, ineffective supervision and poor communication between ECRs and senior scientists and extra pressure to network and publish. Despite the challenges, we found that ECRs consider interdisciplinarity beneficial for both their careers and tackling wicked problems at the coast. We provide a set of practical solutions that acknowledge the need for increased institutional support and change in the system while recognising the role each of us plays in the way we work in interdisciplinarity. We propose five actions to be taken on both an individual and institutional level: (1) accept uncertainty and ensure flexibility, (2) be humble and open-minded, (3) support and lead, (4) think long-term and (5) be patient. These recommendations aim to increase national capability in interdisciplinary research, secure better outcomes for IDCR and provide solutions to wicked problems, and thereby improve the experience of ECRs. While our recommendations are directed to IDCR in the UK context, we believe they provide pathways and ways forward that hold value for interdisciplinarity elsewhere.

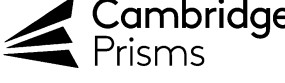

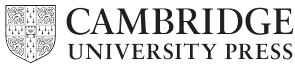

## Introduction

Coastal areas are dynamic environments hosting diverse ecosystems and acting as hubs for economic development. Human activities and growing populations (Reimann et al., 2023) have profoundly altered the planet, leading to the triple crisis of climate change, biodiversity loss and pollution (Passarelli et al., 2021). These threats are particularly felt in coastal regions (Harley et al., 2006; Doney et al., 2012; He and Silliman, 2019) where the latest Intergovernmental Panel on Climate Change (IPCC) report (IPPC, 2022) projected a tenfold increase in coastal flood damages by the end of the twenty-first century and considers it an "existential threat for coastal communities and their heritage" with cascading effects that include the loss of and damage to coastal ecosystems and their services, infrastructure, livelihoods, economic sectors and human health. In response to these global threats, organisations such as the IPCC (2022) and the Organisation for Economic Co-Operation and Development (OECD, 2019) call for urgent cooperative action to achieve effective adaptation and mitigation that will warrant a resilient and sustainable future. United Nations (UN) initiatives like the UN Ocean Decade (Arbic et al., 2024) and the 17 Sustainable Development Goals (SDGs), which are at the heart of the UN 2030 Agenda for Sustainable Development (Moallemi et al., 2020), highlight the need to work across sectors and adopt interdisciplinary approaches. To tackle the complex challenges of climate change and sustainability in coastal systems, it is essential to seek research-based solutions that integrate natural and social sciences and bridge disciplinary and sectoral boundaries (Liu et al., 2007; Renn, 2021; Schipper et al., 2021).

Academic disciplines provide a framework, a shared language and a theoretical background within a field that allows science to progress. However, it is increasingly recognised that challenges involving both human and natural systems, that is, wicked problems, require solutions that span disciplines (Liu et al., 2007; Renn, 2021). Terms such as multidisciplinarity, interdisciplinarity and transdisciplinarity, first introduced in the 1970s (Klein, 2017), have gained prominence over the past three decades, becoming established concepts within academia (Morillo et al., 2003; Porter and Rafols, 2009) and in wider contexts, including the implementation of the SDGs (Moallemi et al., 2020) and risk management (Peek and Guikema, 2021). However, various definitions of these terms exist, often leading to misinterpretation and confusion (Lawrence, 2010; Klein, 2017).

The three terms refer to an increasing level of integration and convergence of knowledge across different domains. The boundaries between disciplines are well established in multidisciplinarity, with disciplines running in parallel, are blurred in interdisciplinarity and transcended in transdisciplinarity (Stock and Burton, 2011; Turner et al., 2017; Mayes et al., 2023). Interdisciplinarity describes research moving beyond the traditional siloed approach with different disciplines working together (Stock and Burton, 2011; Kelly et al., 2019; Peek and Guikema, 2021; Mayes et al., 2023). Here we have chosen to focus on interdisciplinarity, which crucially requires the integration of data, knowledge, theory and methods, and we understand it as a stage before transdisciplinarity (which would further require the inclusion of non-academic actors and/or the synthesis of a new discipline). However, we recognise, as Stock and Burton (2011) point out, that "*boundaries between interdisciplinary and transdisciplinary projects are thus diffuse and dependent more on a subjective judgment on the level of holism applied than on the presence of clear boundary markers*" (p.1102).

Interdisciplinarity may provide a way to tackle complex challenges, but it is also a challenge in itself. Due to its nature, interdisciplinarity requires additional time, people and effort (van Helden et al., 2024). As the emphasis on and interest in interdisciplinary work grows, early-career researchers (ECRs) are often at the forefront, as they are commonly allocated a larger proportion of project time to dedicate to research and output delivery compared to their mid and late career colleagues (Rhoten and Parker, 2004; Haider et al., 2018; Pannell et al., 2019). The definition of ECR varies depending on the discipline, country, university and even department. UK Research Councils, funding bodies and learned societies often base their definitions in terms of years since completion of doctorate (e.g., 10 years for the American Geophysical Union) or first permanent position (e.g., 8 years for the Challenger Society for Marine Science).

Systemic issues in academia, such as field-specific funding, discipline jargon, short-term contracts, lower success in funding for interdisciplinarity (e.g., Rhoten and Parker, 2004; Roy et al., 2013; Bromham et al., 2016) and increasing competition for limited funding (Drakou et al., 2017; Daniel et al., 2022), are amplified for ECRs. This is due to their limited experience and fragile position within academia (e.g., scarce access to networks, short-term contracts, time or funding) (Haider et al., 2018; Hein et al., 2018; Fam et al., 2019; Andrews et al., 2020; Schrot et al., 2020; Rölfer et al., 2022). The increase in the number of ECRs, while funding availability and job opportunities have not kept pace (Maher and Sureda Anfres, 2016), makes the academic landscape more competitive now compared to 20–30 years ago (Fang and Casadevall, 2015; Maher and Sureda Anfres, 2016). Furthermore, interdisciplinary projects often require additional efforts in project coordination, which may decrease productivity, since less time is allocated to research and working on scientific publications (Schrot et al., 2020). This increased competition and the changing expectations with increasing demand for inter- and transdisciplinary approaches imply higher risks for ECRs than more senior, established colleagues since it has the potential to inhibit career advancement (Hein et al., 2018; Fam et al., 2019; Andrews et al., 2020).

National research funders and institutional relationships shape interdisciplinary research by creating funding schemes and providing additional training infrastructure (Lowe and Phillipson, 2009; Lyall et al., 2013). In the United Kingdom, interdisciplinarity is a goal for UK Research and Innovation (UKRI), aiming to push frontiers and deliver impact. This is exemplified by recent programmes such as the Sustainable Management of UK Marine Resources (SMMR) and the Resilient Coastal Communities and Seas programmes, along with their associated networks (SMMRnet and the Coastal Communities and Seas Together for Resilience (COAST-R)), which encourage systems approaches to tackle coastal problems (as identified by policymakers, funders, policy, academics and other practitioners). Such programmes and networks also aim to improve national capacity in inter- and transdisciplinary research in coastal and marine areas. Around 36% of the UK population lives within 5 km of the coast (EC, 2010). In addition, each year over 270 million people visit the coast (Elliott et al., 2018), generating over £13.7bn in tourism spend in England alone (NCTA, 2013). At present, there are 2.4 million properties at risk of flooding from rivers and the sea in England alone (EA, 2025); this number will increase to 3.1 million between 2036 and 2069. Assuming current levels of adaptation and low-end scenarios for population and emissions increases, the annual direct damages to properties are expected to double by 2050, reaching 120 million in England (Sayers et al., 2022).

The experience of ECRs working in interdisciplinary research can differ depending on the country's academic and funding settings. Previous studies exploring the ECR interdisciplinary experience in environmental research have focused on Norway (Deininger et al., 2021), Australia (Blythe and Cvitanovic, 2020; Brasier et al., 2020), Canada (Andrews et al., 2020), New Zealand (Pannell et al., 2019), the United States (Rhoten and Parker, 2004; Benson et al., 2016; Finn et al., 2022) or the global community (Drakou et al., 2017; Haider et al., 2018; Hein et al., 2018; Daniel et al., 2022; Rölfer et al., 2022). However, there is limited research on the experience of ECR in the United Kingdom working in Interdisciplinary Coastal Research (IDCR).

In this paper, we address that gap by exploring the challenges, barriers and opportunities faced by ECRs engaged in IDCR in the United Kingdom. Additionally, we consider whether and how interdisciplinary research benefits (or hinders) ECR career progression. Finally, we provide recommendations on how IDCR practices could be improved to secure better solutions to current coastal challenges while also improving the ECR interdisciplinary experience.

## Approach

This paper writes up the findings of a series of self-organised bottom-up activities initiated and carried out by the ECRs working within the "Resilient Coasts: Optimising Co-Benefit Solutions" (Co-Opt) project at the time of writing. Co-Opt was part of the SMMR programme funded by UKRI (2021–2025). Co-Opt focuses on scalable and adaptive approaches to support coastal and shoreline management and, along with other projects on the SMMR programme, is highly collaborative in its approach, addressing gaps between science and policy identified by policymakers.

As ECRs ourselves, we focused our analysis on the experience of ECRs who have worked or are working in IDCR projects in the United Kingdom. For this study, we define ECRs as researchers who have completed their PhD, have <10 years of research experience and have not yet reached a full level of independence and funding income. The authors include coastal scientists, an environmental scientist, an environmental economist and an environmental social scientist, all of whom have diverse experiences in interdisciplinary research and represent six different nationalities. English is the first language for only one ECR. This paper presents findings gathered through a multimethod approach consisting of three modes of data collection: (1) internal workshops, (2) SMMR webinar and (3) online survey.

First, in an approach similar to that employed by Deininger et al. (2021), we drew on insights from our own experiences as ECRs involved in a large interdisciplinary project focused on coastal management in the United Kingdom. We organised three 1-h internal workshops, attended exclusively by us (authors of the paper), to capture personal reflections on the challenges and opportunities of IDCR. We used a Miro board (www.miro.com) to record and organise our ideas along the three major themes: (1) barriers and difficulties to the increased adoption of interdisciplinarity; (2) causes of these barriers; and (3) solutions, recommendations and reflections. To contextualise our personal reflections and to compare them with the body of literature, as part of this first activity, we performed a literature review. All of the authors contributed to the literature review. Search terms were agreed collaboratively and included "interdisciplinarity", "environment", "coast", "marine", "early career" and "climate". The literature review allowed us to create a list of challenges, barriers and possible solutions as identified by other national and international research in this area.

The output from the workshops was a presentation "Interdisciplinarity in Coastal Research," which was delivered on 6 March 2024, through SMMR-Net online live webinar, open to anyone who registered. To amplify the reach of the webinar, it was advertised through various channels, including social channels and newsletters (i.e., SMMR network, our institutions and our own personal channels and networks). During the webinar, in addition to question-and-answer sessions and general discussions, we conducted several polls using Poll Everywhere (https://www.polleverywhere.com/) to collect data about participants' experiences. These polls included questions aimed at discovering potential solutions for improving the ECR experience in IDCR, as well as benefits associated with such research (see Supplementary Materials). These questions were built upon our internal workshop and aimed to expand our understanding of the IDCR experience beyond the limits of the Co-Opt project. The live webinar brought together 31 participants from universities, government agencies and industries across the United Kingdom. The number of responses to the polls varied from 11 to 20, depending on the question. Attendees represented various disciplines, including coastal modelling, sound art/audio research, science communication, marine biology, marine microbiology, social sciences and coastal remote sensing.

Finally, to explore the perceptions and experiences of a broader group of ECRs, we conducted an online survey hosted on Qualtrics XM targeting ECRs in the United Kingdom. The survey was approved by the Ethics Committee of the School of Geography and Sustainable Development at the University of St Andrews (approval code GG18066). We recruited participants, working on or interested in IDCR, through relevant academic newsletters and social media platforms. The survey consisted of 30 questions, mostly multiple choice (see Supplementary Material). Participants had the option to skip questions they did not wish to answer. The survey was built on the two previous activities (internal workshops and webinar) and aimed to further deepen our understanding of IDCR in the UK context. Participants were asked to identify and rank the most significant barriers to IDCR, the underlying causes of these barriers and the opportunities IDCR presents as identified during the previous activities (i.e., internal workshops and webinar). Open-ended questions gave participants the opportunity to further reflect and share their experiences and ideas. After excluding empty or incomplete surveys (those that did not meet the 70% complete threshold) from 63 total recorded responses, we received 20 valid responses.

The answers to open-ended questions from the online survey and polls, and information captured in webinar discussions and internal workshops, were qualitatively analysed using Microsoft Excel assigning codes that emerged from the data itself and the literature (see Supplementary Material for the coded excerpts). Initially, we analysed the results of each method separately, and the preliminary results of each activity informed the following activity (please see above). Given the consistency of results across all activities, we decided to treat data collected from the three methods as a single dataset. The full qualitative analysis presented here was carried out on the complete data set. Each response was assigned a code and a predefined broad category – systemic issues, disciplinary differences or project structure. The categories were identified in the internal workshops, with systemic issues previously highlighted by Deininger et al. (2021). Systemic issues are those ingrained within the academic and funding systems.

Disciplinary differences occur due to different norms and methodologies between academic fields. The project structure category was used for those responses related to the way projects are designed. For the qualitative analysis, we ran a series of collaborative online sessions where we reviewed each of the data excerpts. Each of us individually assigned a code and a category to each data excerpt. If our individual codes were consistent, we kept them as a code at this stage. If we disagreed, we discussed and tried to reach an agreement. Once we finalised this first step of coding, we then reviewed the codes and categories to ensure a coherent set and remove any repetitions. An additional session allowed us to verify, refine and consolidate the final codes and categories and agree on data visualisation. The quantitative survey data were explored with data visualisation methods and descriptive statistics using SPSS v.29.0.

## Results

Findings were generally consistent across the different data collection methods used in this study. Consequently, we present the findings in an integrated manner, highlighting the method only when the collected data pertains to a particular method.

### Online survey respondent profile

We used the online survey to collect more detailed information and gather perceptions of a larger group of respondents, which was not possible during the webinar. To ensure active participation in the discussion, webinar participants were only asked a limited number of questions. Below, we present additional information collected exclusively from the online survey.

Most survey participants were between 25 and 34 years old ($n = 10$, 50%) and the majority were female ($n = 13$, 65%). Less than half of the respondents (45%) had fixed-term contracts (for a set duration), the other half had open-ended (permanent) contracts and one respondent was a freelance researcher. Over half had research-only focused contracts (60%). All but one respondent identified themselves as ECRs. The respondents were on average 3.18 years post-PhD completion, ranging from just finished to 8 years. Nearly all respondents (90%) described themselves as interdisciplinary researchers. However, only 35% stated interdisciplinary research as their main field of study. For 30% the main broad research field was natural sciences (i.e., marine biology or marine ecology), for 20% it was physical sciences (i.e., physical oceanography or coastal oceanography) and for 10% it was social sciences, including sustainable development. Most respondents' work spanned both marine and coastal habitats (45%). The majority (85%) were or had been involved in interdisciplinary research projects focusing on coastal and marine habitats and/or communities. Two respondents had not been involved in such projects but expressed interest in working in interdisciplinary projects in the future and explained that their involvement had been limited by "*money*" and "*lack of deep knowledge and lack of evidence to prove my ability in this field.*" Four respondents were part of the SMMR-funded projects, with one respondent simultaneously working on another interdisciplinary project. Four respondents were part of other interdisciplinary networks, such as COAST-R, Surface Ocean – Lower Atmosphere Study, HORIZON EU ECR network and university research groups. In terms of mobility post PhD, 50% of the respondents had worked for a sole institution, 15% for two institutions, 15% for three institutions and 10% for four institutions; only one respondent had remained in the same institution after they completed their PhD.

### Barriers and causes

The qualitative analysis highlighted that the most recognised barriers were systemic in nature. These included barriers linked with employment, such as short-term contracts, power dynamics and lack of professional development opportunities (Figure 1). Barriers such as demanding workload, limited institutional support, ineffective supervision and communication between ECRs and senior scientists and extra pressure to network and publish were recognised to be the most widespread challenges. On the other hand, finding suitable journals and the lack of recognition of interdisciplinarity were considered the least prevalent challenges (Figure 2). Survey respondents themselves had experienced demanding workload (50%), a lack of jobs and the highly competitive nature of academia (50%), a lack of available and suitable funding (50%), short-term contracts (45%) and stress due to the extra pressures to network and publish (40%). The issue with short-term contracts was highlighted by this participant: "*losing time for applications for jobs, the stress of feeling disposable. Short-term contracts are a huge fault of the research system that fail a lot of talented young researchers eventually pushing them out of research.*"

Disciplinary differences were also highlighted as often hindering IDCR. These included different epistemologies, cultural differences and a lack of appreciation of social scientists (Figure 1). The internal workshops highlighted different vocabularies and different domains of knowledge (due to differing foundational concepts, questions and assumptions, as well as deep subject specialisation) as barriers to successful interdisciplinary collaborations. For example, coding for social scientists involves adding a descriptive label to data extracts, while physical scientists understand coding as writing programming scripts. Language barriers between different disciplines were also among the top five barriers as ranked by survey respondents (Figure 2). Webinar attendees also identified challenges such as impostor syndrome "*…that people feel like disciplinary outcasts, that kind of rang with me. […] adapting learning new things every day, you kind of feel like an outcast…*" and fatigue from being an advocate of interdisciplinary research "*… there is kind of a… fatigue if you are an interdisciplinary individual having to always be the voice of the underrepresented disciplines.*"

Other issues identified were at the project structure level. The way that projects are designed and funded may not consider the timelines required, even when the funding is specifically aimed at achieving interdisciplinary outputs. Interdisciplinary work depends on data from varied sources (due to the wide range of ways in which these data are produced), which can arrive on mismatched timescales, causing delays to the project overall.

Although the survey focused mostly on barriers and solutions, we asked the survey respondents to identify and rank the main causes of these challenges. The most prevalent causes were identified to be the lack of experience of senior scientists/Principal Investigators in interdisciplinarity, a lack of sufficient funding for training and personnel and time-consuming procedures (Figure 3). Participants highlighted that there is a lack of support for ECRs' career development, for example, through the recruitment of PhD students. Participants highlighted power dynamics as another cause and indicated that their voices are "*vulnerable to lack of respect from authority, and open communication is often seen as arrogant.*" It was also acknowledged that there is a lack of understanding of what interdisciplinary research entails: "*Interdisciplinary is like hybrid,*

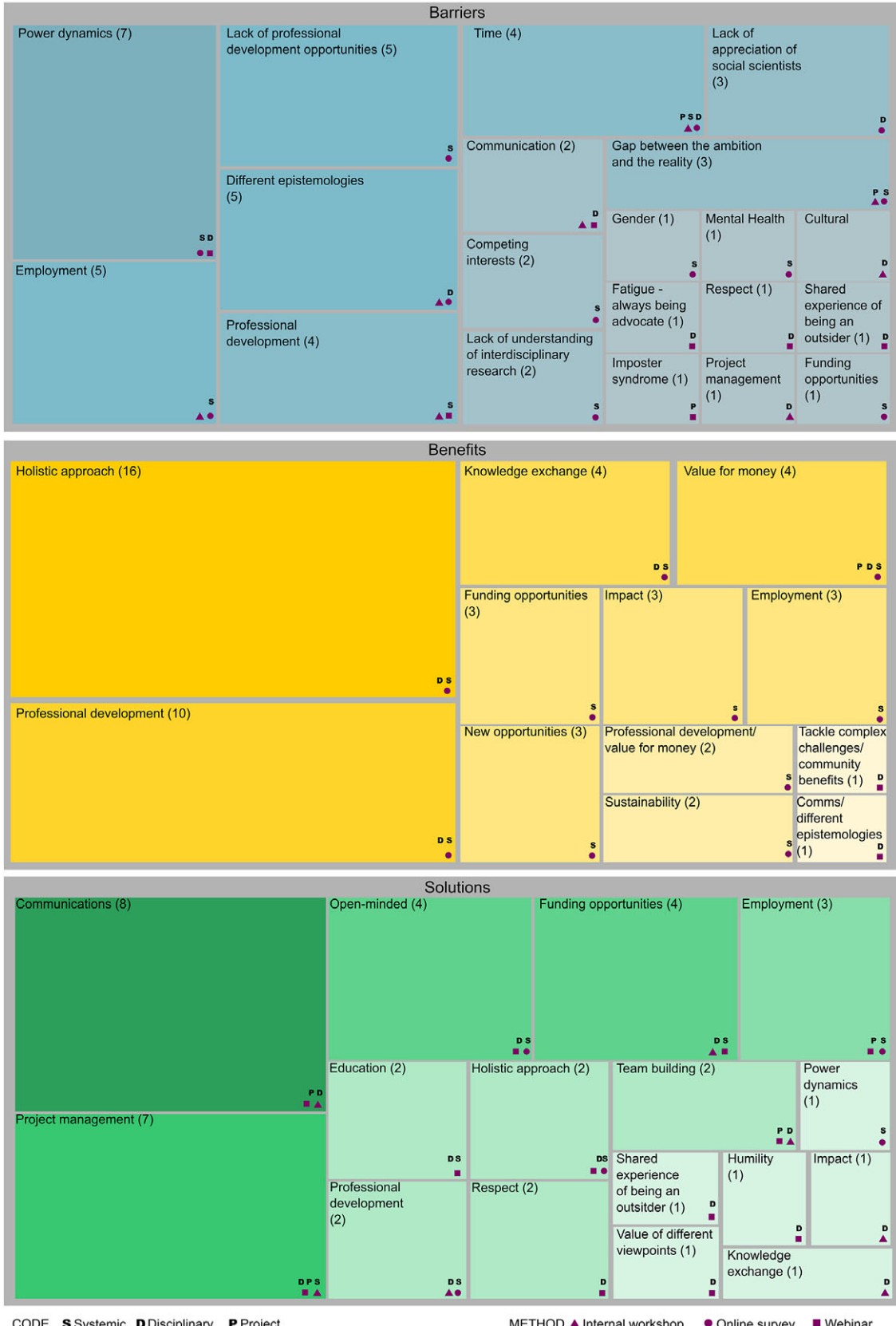

**Figure 1.** Codes assigned during qualitative analysis for the combined data. The analysis is performed for barriers, benefits and solutions. The size of the rectangle and the number between brackets indicate the number of times the code is registered. Codes are classified in systemic, disciplinary and project-related. Whether it was recorded in the internal workshops, the online survey or the webinar is indicated by a triangle, circle and square, respectively.

## Barriers

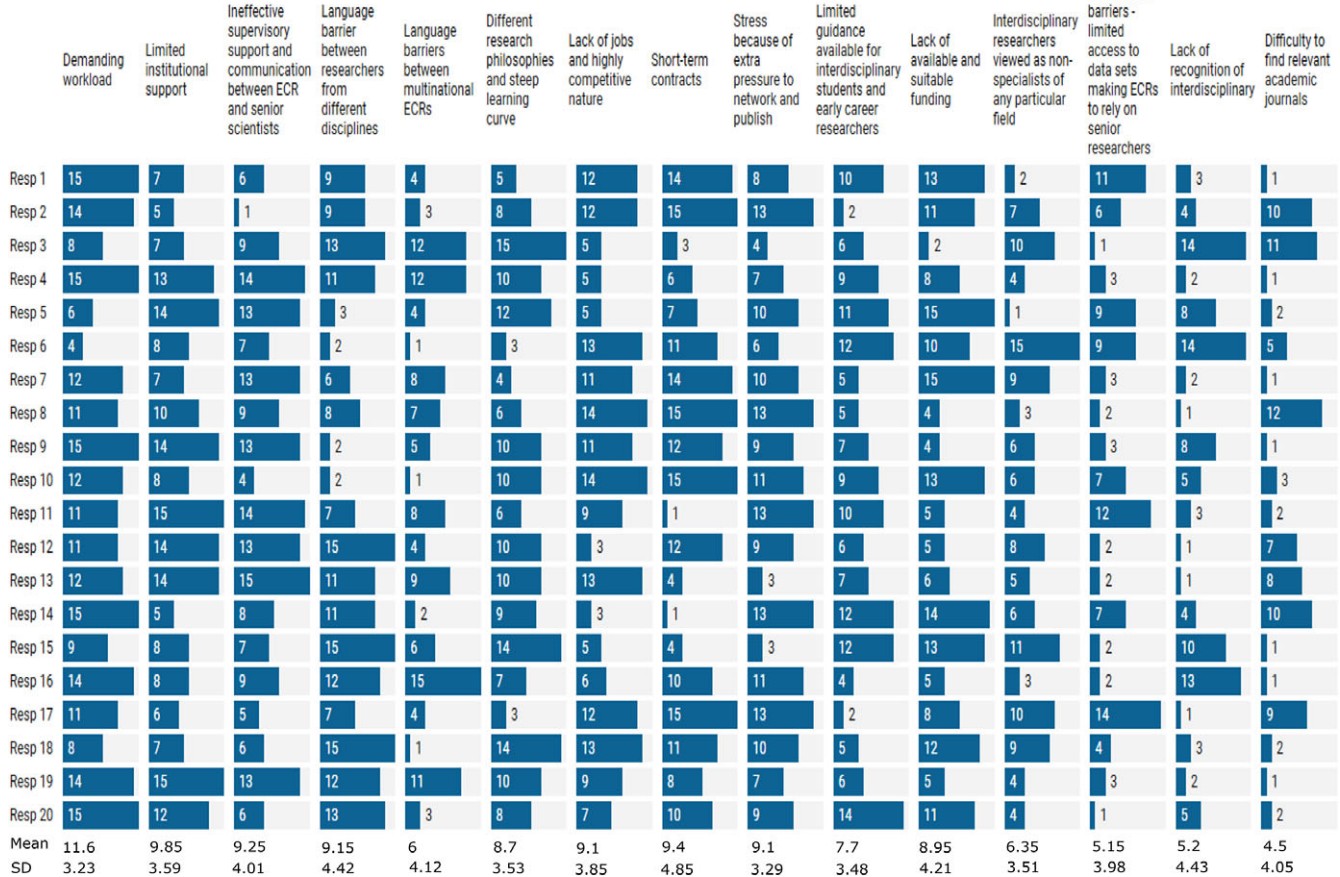

**Figure 2.** Ranking of the barriers to IDCR experienced by ECRS. 15 = most prevalent barrier, 1 = least prevalent barrier. Mean values and standard deviations were calculated for each barrier.

rare and unique unfortunately not most of the market demand understands this concept. I'd say the society needs the change of mindset."

### Impacts for research and career progression

Despite various challenges, overall, ECRs felt positive about the benefits interdisciplinarity brings to coastal research. Our qualitative analysis identified the ability to provide a holistic approach ("*It conveys the whole picture, the marine environment is not on its own*"), professional development ("*…it helps to upskill oneself […] while giving optimum benefits to the society and environment*"), knowledge exchange ("*expansion of data resources, capacity building, increased knowledge and more outputs*") and value for money ("*Increases efficiency of funding for tackling issues/questions working together instead of separately*") as some of the main benefits (Figure 1). Survey respondents highlighted that such research is more holistic, accounts for the complexity of issues, has a greater impact and fosters knowledge exchange. Additionally, most of the respondents (80%) reported that they believe that IDCR is also beneficial for their career development. The explanations included that it "*gives more transferable skills. Improves ability to think flexibly and out-of-the-box,*" it provides "*more job opportunities (it's easier to change discipline if needed)*" and it "*opens doors to opportunities and is a lot more meaningful (less siloed thinking).*"

### Solutions for improved interdisciplinary practice

The qualitative analysis (Figure 1) highlighted solutions related to communication, project management, open-mindedness and funding opportunities. Participants prioritised solutions (Figure 4) at the project level. The top solution was identified to be joint leadership of task and work packages to include people from different disciplines (e.g., one from social sciences or humanities and one from natural or physical sciences). This may ensure better integration of perspectives and tasks throughout the project ("*Map different objectives from each of the disciplines and prioritize based on a timeline following the overall goals*"). Greater flexibility in approach and methodologies and contingency plans were also highly ranked as possible solutions ("*Be honest with timescales and capabilities*"). Another solution highlighted in the surveys was the figure of an "*interdisciplinarity champion within a project.*" This would be a person with previous experience working in interdisciplinarity, with the skills to communicate across disciplines and bring people together. Other highly rated suggestions included increased length of project funding and more frequent in-person meetings.

When it came to solutions for disciplinary differences, in our internal workshop, we acknowledged that none of us can be expected to know about everything. However, it is vital for communication that we all understand the basics of each other's work as highlighted by the participants in the webinar "*Creating a framework with unified terminology and methods.*" This can be achieved through taking the

## Causes

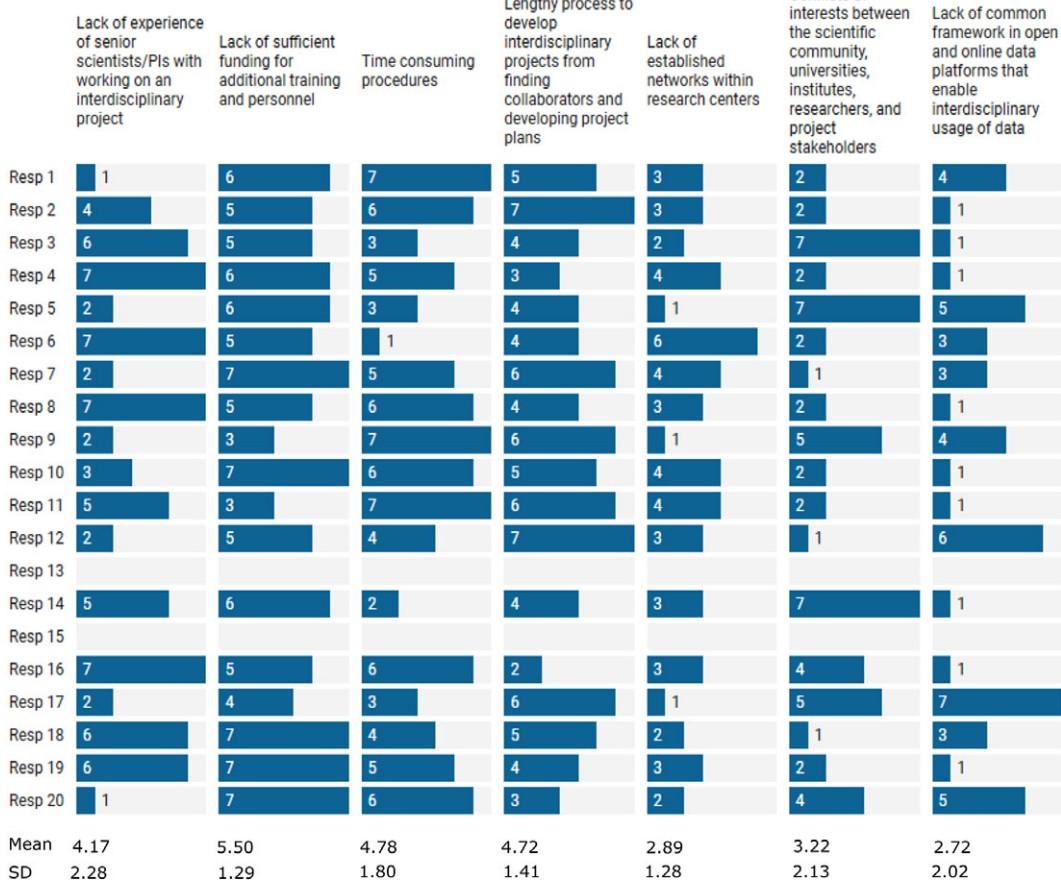

**Figure 3.** Ranking of the causes of IDCR barriers experienced by ECRS. 7 = most prevalent cause, 1 = least prevalent cause. Mean values and standard deviations were calculated for each cause.

time to engage across the disciplines and to share your work, including information on specific methodologies, vocabularies and so forth. We need to ensure that we take enough time to reflect and understand each other, which may require additional meetings as highlighted by the survey participants, "R*egular meetings between PDRAs [postdoctoral research associates] and others to talk, reflect and understand each other.*" Survey respondents recognised regular meetings as one of the main solutions (Figure 4). Additional training and networking opportunities may be useful in gaining an understanding of different disciplines and ways of working.

Although most of the solutions proposed changes to ways of working and project structures, solutions to systemic issues were also provided. It was suggested there may be a need for funders to learn from funding programmes such as SMMR and to reflect on learnings for future funding rounds. Additionally, the availability of permanent academic and research positions can enhance the productivity and reduce the stress of ECRs. For example, finding a permanent position was indicated to be a turning point that allowed a respondent "*to stop losing time for applications for jobs and focus on my research, plus it gave me the enthusiasm to invest in my research community without the stress of feeling disposable.*"

### Discussion

Coastal systems inherently require the involvement of various disciplines because they exist at the intersection of land, sea and atmosphere, encompassing natural, physical and social processes. Therefore, interdisciplinarity in coastal research has been embraced worldwide, evidenced by a number of projects (e.g., CHERISH, n.d.; InterFACE, n.d.; ICAP-2, n.d.; SeaLex, n.d.). The complexity of wicked problems and the need for interdisciplinary research to tackle them prompt many ECRs to seek roles in interdisciplinary projects (Spence et al., 2024). ECRs are often responsible for the delivery of the core tasks of interdisciplinary projects. However, their aspirations may be compromised by the many challenges and barriers they face. While previous research (e.g., Drakou et al., 2017; Hein et al., 2018; Pannell et al., 2019; Andrews et al., 2020; Blythe and Cvitanovic, 2020; Deininger et al., 2021) has focused on other geographical areas and in environmental sciences, our focus is on the experience of ECRs in IDCR in the context of the United Kingdom. This is especially relevant given the significant risk of coastal flooding and erosion (EA, 2025), and as the institutional context is crucial in shaping scientific research (Lyall and Fletcher, 2013).

### What barriers do ECRS face in conducting IDCR in the United Kingdom?

One of the biggest challenges for ECRs involved in IDCR was reported to be the demanding workload. Andrews et al. (2020) have suggested that ECRs tend to overcommit either by necessity or choice in collaborative settings. Demanding workloads are a

## Solutions

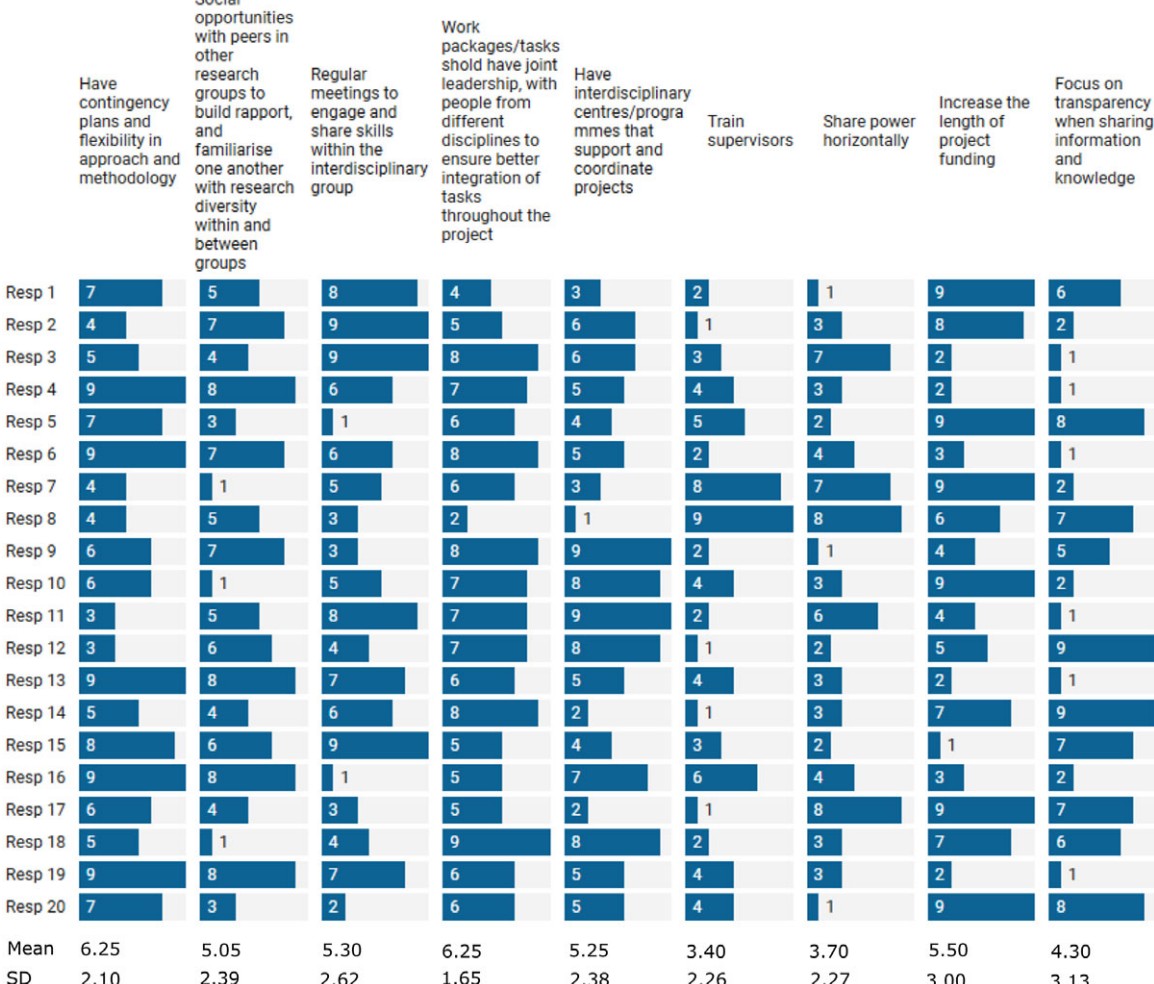

**Figure 4.** Ranking of the solutions for successful IDCR suggested by ECRS. 9 = most prioritised solution, 1 = least prioritised solution. Mean values and standard deviations were calculated for each solution.

widespread issue for ECRs in general, not only for those involved in interdisciplinary research (Susi et al., 2019; Humphries et al., 2021; Christian et al., 2022). The early years after PhD completion are a pivotal stage in the career of researchers, hoping to pursue an academic career. Researchers are building their professional reputation by increasing their publishing records and applying for grants while often also contributing to teaching and administrative responsibilities (Aprile et al., 2020; Smithers and Gibbs, 2024). For those involved in interdisciplinary research, these demands can be exacerbated by additional administrative requirements and more competitive (and limited) funding opportunities (Deininger et al., 2021). Furthermore, this is often under the stress of short-term contracts and a lack of relevant job opportunities, recognised in this and previous studies on interdisciplinary research (Hein et al., 2018; Deininger et al., 2021). These factors put pressure on ECRs who often express concerns about their work–life balance (Brasier et al., 2020). The pressure to publish to maintain a role in academia and advance to more senior positions can cause stress, anxiety and depression (Andrews et al., 2020; Cilli et al., 2023). However, in our study, mental health and well-being were only mentioned by one respondent. This does not, however, suggest that it is not an issue among ECRs in the United Kingdom, and is

more likely due to the data collection approach and the relatively small sample size.

Another major barrier recognised by ECRs is limited institutional and ineffective supervisory support, also found by other authors (Morse et al., 2007; Blythe and Cvitanovic, 2020), even though supervisors and mentors report being more aware of their role in shaping interdisciplinary research (Kelly et al., 2019; Andrews et al., 2020; Shellock et al., 2022). In the United Kingdom, the value of interdisciplinary research is widely endorsed by the higher education institutes and funders (Evis, 2021), resulting in interdisciplinary funding programmes. For example, the SMMR (n.d.) and COAST-R (n.d.) networks coordinate interdisciplinary and transdisciplinary research projects that address coastal and marine challenges with significant policy implications. The study participants urged the funders to consider the evaluation and lessons learned from existing programmes to inform future funding calls and interdisciplinary research strategies at the national and international level (Lyall and Fletcher, 2013; Carr et al., 2018).

Despite growing interest in interdisciplinarity, there are still gaps in formal training provision within higher education, with only 20% of UK institutions reported to provide specific interdisciplinary programmes (Evis, 2021). The results of this paper

indicate a gap between strategically supporting interdisciplinary research and offering practical support to ECRs and equipping senior researchers with adequate mentorship skills. This is not unique to IDCR in the United Kingdom, discrepancies between what is expected by researchers and the support provided by the academic system have been noted to occur in other fields, such as interdisciplinary biomedical research (X-Net, 2024) and in other locations (e.g., Australia (Newman, 2023)).

ECRs can be confronted with other, more practical barriers when asked to work outside of their knowledge zone. Each discipline has its own distinctive "language" and ECRs need to acquire new skills and become familiar with differences in terminology, something that requires additional time (Andrews et al., 2020; Deininger et al., 2021). Researchers across disciplines may be using the same words to mean subtly different things and are not always aware of these differences, resulting in talking at cross purposes. Differences in the method of communication between disciplines may also lead to misconceptions because of the complexity of the different types of knowledge (Bracken and Oughton, 2006; Dick et al., 2017; Kelly et al., 2019). However, this can be addressed by creating a shared glossary and holding regular meetings. Further issues with data collection, analysis and interpretation may arise because of the differences in the methods and approaches between physical and social sciences (Palmer et al., 2016; Finn et al., 2022; Shah et al., 2023). Data barriers may refer to the difficulties of data dissemination between different disciplines, the various formats and issues of accessibility, as ECRs often rely on senior scientists and institutes for data accessibility (Deininger et al., 2021). Delays in receiving data can leave tasks in limbo, preventing researchers from progressing with their work. This is especially likely when the time required to complete tasks is underestimated during planning due to unfamiliarity with other disciplines.

Finally, an important barrier that is often overlooked is the language barrier. Worldwide, English remains the primary language of scientific research (Amano et al., 2016). However, in modern science, interdisciplinary work is carried out by multiple nationalities. The prevalence of English may hinder networking and the expression of ideas, and may eventually result in limited research output for non-native speakers (Amano et al., 2016; Brasier et al., 2020; Deininger et al., 2021). Our personal experience suggests otherwise. In our team, all but one were non-native English speakers, which made us more conscious of the language used and led us to explain our thoughts without relying on disciplinary jargon.

### How does being part of an interdisciplinary research project impact ECRs' career progression?

The survey and webinar results reinforced our hypothesis that IDCR is essential for broadening our understanding of coastal systems and solving complex issues. We also found that most survey respondents believed that the experience of interdisciplinary research has a positive impact on their careers. However, this could be due to self-selection bias as the participants were not selected randomly but voluntarily chose to participate in the online survey or attend the webinar. Other studies outside the United Kingdom have reported ECRs to be discouraged from participating in such projects because of concerns that this may undermine their career development and tenure goals (Hein et al., 2018; Mäkinen et al., 2024). On the other hand, Millar (2013) found that PhD graduates with interdisciplinary dissertations in Australia were more successful in securing academic positions and published more papers. This indicates that while there might be no shortage of fixed-term positions on interdisciplinary projects, disciplinary boundaries come into play when securing permanent positions. This is often reinforced by academic structures and senior academics who often "maintain the institution of disciplinary order by creating conditions and engaging in practices that devalue interdisciplinary research in tenure and promotion reviews" (Mäkinen et al., 2024).

Interdisciplinary research is undoubtedly more time-consuming, and time was found to be a major barrier in this study, not only for IDCR but also regarding career progression. The time required to develop collaboration and employ new methods can limit the outputs, increase career uncertainty (Bridle et al., 2013; Kelly et al., 2019; Pannell et al., 2019; Brasier et al., 2020; Schrot et al., 2020) and increase cognitive burden (Park, 2025). Under the "publish or perish" mentality, ECRs often feel that they do not receive adequate recognition for their work, discouraging them from engaging in interdisciplinary research (Rhoten and Parker, 2004; Roy et al., 2013; Benson et al., 2016; Kelly et al., 2019). Yet, there is no consensus on how interdisciplinary research affects publishing records. Previously, it has been reported that interdisciplinary researchers publish less frequently than those not crossing disciplinary boundaries (Leahey et al., 2017; Daniel et al., 2022). This has been linked to difficulties finding the relevant journal and reviewers (Pohl et al., 2015; Daniel et al., 2022; Zhang and Wang, 2024). However, recent research has found that manuscripts with larger knowledge-base interdisciplinarity (measured through references) are associated with higher acceptance rates than manuscripts with higher topic disciplinarity (measured through title and abstract) since the former demonstrate a broader knowledge of the literature (Xiang et al., 2025).

There is also a perception that interdisciplinary research is often held in lower regard by colleagues within disciplines (Hein et al., 2018). Those identifying themselves as interdisciplinary researchers can feel like they "work twice as hard" but "get half the credit" (X-NET, 2024). However, this is something that might not always be spoken about openly. In our study, this was only raised in the webinar discussion, highlighting that safe spaces (Bridle et al., 2013) and qualitative research can give a more detailed understanding of the experience of those involved in interdisciplinary research. Working in an interdisciplinary manner aims to cover knowledge areas within a project that are not held by all to address complex challenges that require collaboration and co-learning.

### Conclusion and recommendations

Despite facing multiple challenges, ECRs strongly acknowledge the benefits of interdisciplinary research for addressing complex problems and for their own professional development. The current overall research and funding landscape in the United Kingdom encourages interdisciplinary research. However, our research highlights a mismatch between the ambition and the day-to-day reality of ECRs working in IDCR and the need for increased institutional support that acknowledges and fosters the diversity and experience of ECRs. While our findings aim to be representative of the UK workforce in IDCR, we are mindful that it represents the views of a small group of ECRs. We do not claim that our findings are exhaustive, but we do believe they illustrate ways forward for IDCR. The following recommendations aim to secure better outcomes for coastal areas by improving IDCR practices and enhancing the experience of ECRs.

1. **Accept uncertainty and ensure flexibility:** Interdisciplinary research can be unpredictable; therefore, it requires flexibility from the project team, the steering group and the funder. This

requires adjusting the funding system for interdisciplinary research.

2. **Be humble and open-minded:** Every new interdisciplinary project will involve a learning curve, as the mix of disciplines and personalities will be different. Use the opportunity to learn from each other, organise regular meetings and workshops.

3. **Support and lead:** Interdisciplinary programmes and networks, such as SMMR and COAST-R, should be seen as an opportunity to champion new leaders who are more aware of interdisciplinary working methods. Support interdisciplinary networks and give voice to ECRs.

4. **Think long-term:** While some challenges can be tackled at the individual level, systemic changes are also needed. A strong long-term vision that learns from successful programmes and sustained strategic funding that supports interdisciplinarity is encouraged.

5. **Be patient:** Interdisciplinarity demands more resources and requires longer timescales for fruitful scientific research and successful outputs. Take time to develop strong long-term collaborations, evaluate the process and adapt as needed.

**Open peer review.** To view the open peer review materials for this article, please visit http://doi.org/10.1017/cft.2025.10022.

**Supplementary material.** The supplementary material for this article can be found at http://doi.org/10.1017/cft.2025.10022.

**Data availability statement.** The data that support the findings of this study will be openly available on Zenodo upon paper acceptance (https://doi.org/10.5281/zenodo.17938260).

**Acknowledgements.** The authors would like to express our gratitude to the survey respondents and webinar participants. The authors would also like to thank Chrissy Onay for inviting us to speak at the SMMR webinar series.

**Author contribution.** EA initiated the idea of the workshops in preparation for the webinar. All authors participated in the internal workshops and participated in the webinar. EA and MPP curated the online survey. EA, MPP, AB, MM and CM conducted the analysis. EA and MPP led the writing of the manuscript. EA, MPP, AB, MM, CM and SK reviewed and edited the manuscript.

**Financial support.** This work was conducted within the "Resilient Coasts: Optimising Co-Benefit Solutions" (Co-Opt) research project funded through the NERC-ESRC Sustainable Management of Marine Resources Programme (NCR10332), NE/V015532/1 (University of Liverpool), NE/V016423/1 (National Oceanography Centre), NE/V016245/1 (University of St Andrews) and NE/V016490/1 (Cranfield University).

**Competing interests.** The authors declare none.

**Ethics statements.** The survey was reviewed and approved by the Ethics Committee of the School of Geography and Sustainable Development at the University of St Andrews, approval code GG18066.

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
