## [Reviewer Report]

Many thanks for the opportunity to review this paper. ECRs are a crucial part of the coastal research community and it is so useful to see these insights drawn from the community. While the challenges raised in the paper are perhaps not novel and have perhaps been persistent for some time, there is a real need to continue to raise these issues and ensure the status quo is challenges. I have recommended some relatively minor revisions below, but my hope is that later career researchers will read this paper with interest and recognise their role in being advocates and champions for their ECR colleagues.

6 – Is that certain? It reads as though it is only ECRs who do this work. I wonder if this could be phrased as ‘increasingly, ECRs are undertaking interdisciplinary research, often more frequently than their mid and late career colleagues – this can be a challenge’.

10- could the date of when this research was carried out be included in here?

13 – the solutions could be drawn out a little more in the abstract. Could more of what is in the impact statement be integrated into this, please?

21 – I think these two initiatives probably need to be written in full even in the impact statement.

57 – recommend writing out the Ocean Decade in full here and then refer to it as the Ocean Decade. Also recommend stating that they are the UN SDGS.

61 – is there a citation that could be used here? There are a number of papers which discuss the importance of interdisciplinary solutions – perhaps some of Chris Cvitanovic’s work could be useful here.

71 – given that there is recognition of the different terms in the paper, why is the focus on interdisciplinarity rather than transdisciplinarity, or indeed both? What was the justification for focusing on interdisciplinary work? This links to the point raised later at L99 where inter and transdisciplinary projects are discussed together.

105 – is coastal research a priority for the UK? The paper used here is quite old and I’m not sure many of us working in this space would agree – is there anything that can be cited here to support this?

112 – programs and networks? SMMR is a program while Coast-R is a network – ReCCS is the program.

114 – these calls for projects are not only written by policy-makers – funders, policy, academics and other practitioners feed into the development of these calls so it’s not strictly correct to say the priorities are only those identified by policy.

168 – 20 seems quite a small sample size to represent the UK’s ECR community in this area. Do you know how representative it is?

173 – some more detail on the coding process would be useful here. It states all authors did the coding – but what did that look like? What was the coding process? Were things checked by different reviewers? How many times were codes reviewed?

264 – the positioning of the section on the respondent profile seems a bit strange as it seems to come after the results are presented. Recommend moving this to before the main body of the results so we have insight into who the people involved were before reading their views.

305 (and linked to the text at ~331) – I am surprised that there wasn’t something said about the lack of interdisciplinary undergraduate courses in the UK? We know this is a significant issues and gap in the educational provision. Were any thoughts on this raised? This feels like it would impact the pipeline and also the supply/ demand for interdisciplinary skills?

---

## [Reviewer Report]

Thank you for the opportunity to review this article - it raised several compelling insights, and it was an interesting read. I have outlined several areas of improvement for the article, and more detailed feedback below. I hope that the authors - who are self-described ECRs - see these suggestions and comments as an avenue to improve the paper so that it reaches its maximum impact. In summary, the main areas of work fall in further detailing and justifying the methods. Many of the other comments ask for more clarity / precision in what the authors are trying to convey.

General comments

No figures in text - it was challenging to interpret some of the results without these

Switches between first and third person - e.g., from line 246 it shifts to first person, line 361

98-101. Yes - but doesn’t it also have significant benefits that should be identified too? Increasing interdisciplinarity leadership, which is hinted at here through the recognition that ECRs often ‘do the doing’ for interdisciplinary work through increased project time, is also a significant competitive advantage. This can also be inferred from line 215, and again at 229. Just a suggestion, but I think it could be really interesting to question what this could look like, or what the implications of this may be.

Introduction

Why the focus on interdisciplinarity, and not transdisciplinarity, particularly when in line 71-75 it is said that transdisciplinarity is the most integrated?

Method

140-146. More detail is needed here for these workshops. Were they organised throughout the Co-Opt project to capture evolving experiences and perspectives, or was the focus more on reflection after the project had ended? Was there a specific method for capturing and recording ideas, or a structure to the workshop? Was it organised by the authors only, or was a broader group of ECRs from across the group included? Are there any similar methodologies that can be cited here?

Line 147-157. Was the criteria for attending the SMMR-net webinar engagement with an SMMR project? What were the questions used for the polls in this webinar? Why were they identified - was this a direct result of reflections, gaps or validations identified from the three internal workshops?

Line 158-180. Were the results of the first workshops and survey used to identify the questions for the survey?

Line 166. More information is needed about the rapid literature review. Was this collaborative, by all authors? What search terms were used?

168. Were all questions mandatory? If not, how did you determine if a survey was incomplete?

170-180. Interesting that all data from the multiple methods was analysed together. Were the methods also analysed separately, perhaps to feed into the design of the next method? This would be helpful to add here.

173. I don’t understand the sentence - could this perhaps be explained further?

176. How were these broad categories identified?

179. How did you manage all authors coding the data - how did you ensure that codes were used consistently?

Results

200-202. This is interesting - could you give some examples of these different vocabularies, and further detail what is meant by ‘lack of knowledge of different fields’

214-219. This section feels quite brief, and could be further developed.

217-219. I think these may be two separate points that should be expanded upon.

I would change the heading 3.3 - solutions for what?

237 - joint leadership how? E.g., by disciplines, career stages? I also don’t think the quotes following evidence this finding

242 - that’s a really interesting and valuable suggestion

245- what would increased length project meetings achieve?

Line 253 - unclear acronym

246 -263. I would separate out the paragraph into two, with the second focusing on systemic issues. The insight about the need for funding to learn from project experiences is really interesting, and should be explored more.

270-271. This is interesting. What do you mean by fixed term here

I would move section 3.4 to the start of the results to add better context to the findings.

Discussion

323-325. This sentence seems out of place here

Discussion and conclusion are well written and compelling

---

## [Editor Report]

I think this is a timely and pertinent paper. Although the recommendation is for a minor revision (but noting that one reviewer thought the level of revisions warrented a major revision category), I want to stress that I think it is very important to carefully following the suggestions made by the 2 reviewers - the suggestions have been made in a very constructive, positive and supportive manner to assist ECR authors navigate the challenge of publishing their work. The reviewer suggestions are primarily focussed on ensuring that the methods and results sections have the clarity and robustness to support what are well written and well justified Introduction and Discussion sections. Please make sure that you clearly identify how you have revised the mansucript against each of the reviewer suggestions (both general and detailed).

Thank you for submitting your manuscript to Coastal Futures and I very much look forward to seeing the revised manuscript.

---

## [Editor Report]

Thank you for making the revisions to the manuscript and closely following the advice to resubmit the article. I am happy that you have acknowledged and acted upon the reviewer suggestions and can, therefore, recommend the article is accepted. Please note that there remains a requirement to submit a Title page and a graphical abstract, and the journal can provide assistance if required.